# Non-Viral Systems Based on PAMAM-Calix-Dendrimers for Regulatory siRNA Delivery into Cancer Cells

**DOI:** 10.3390/ijms252312614

**Published:** 2024-11-24

**Authors:** Pavel Padnya, Igor Shiabiev, Dmitry Pysin, Tatiana Gerasimova, Bahdan Ranishenka, Alesia Stanavaya, Viktar Abashkin, Dzmitry Shcharbin, Xiangyang Shi, Mingwu Shen, Anastasia Nazarova, Ivan Stoikov

**Affiliations:** 1A.M. Butlerov Chemistry Institute, Kazan Federal University, 18 Kremlyovskaya Str., 420008 Kazan, Russia; padnya.ksu@gmail.com (P.P.); shiabiev.ig@yandex.ru (I.S.); pysin_dima@mail.ru (D.P.); 2Arbuzov Institute of Organic and Physical Chemistry, FRC Kazan Scientific Center, Russian Academy of Sciences, 8 Arbuzov Street, 420088 Kazan, Russia; tatyanagr@gmail.com; 3Institute of Biophysics and Cell Engineering of NASB, 27 Akademicheskaya St., 220072 Minsk, Belarus; ranishenka@gmail.com (B.R.); alesiastanovaya@gmail.com (A.S.); viktar.abashkin@gmail.com (V.A.); shcharbin@gmail.com (D.S.); 4State Key Laboratory for Modification of Chemical Fibers and Polymer Materials, Shanghai Engineering Research Center of Nano-Biomaterials and Regenerative Medicine, College of Biological Science and Medical Engineering, Donghua University, Shanghai 201620, China; xshi@dhu.edu.cn (X.S.); mwshen@dhu.edu.cn (M.S.); 5CQM—Centro de Química da Madeira, Universidade da Madeira, Campus Universitário da Penteada, 9020-105 Funchal, Portugal

**Keywords:** dendrimers, siRNA, drug delivery, thiacalixarene, self-assembly, toxicity

## Abstract

Cancer is one of the most common diseases in developed countries. Recently, gene therapy has emerged as a promising approach to cancer treatment and has already entered clinical practice worldwide. RNA interference-based therapy is a promising method for cancer treatment. However, there are a number of limitations that require vectors to deliver therapeutic nucleic acids to target tissues and organs. Active research is currently underway to find highly effective, low-toxic nanomaterials capable of acting as nanocarriers. In this study, we demonstrated for the first time the ability of symmetrical polyamidoamine dendronized thiacalix[4]arenes (PAMAM-calix-dendrimers) to form stable positively charged complexes with siRNAs, protect them from enzymatic degradation, and efficiently deliver gene material to HeLa cells. A distinctive feature of PAMAM-calix-dendrimers was the unusual decrease in hemo- and cytotoxicity with increasing generation, while these compounds did not cause toxic effects at concentrations required for siRNA binding and delivery. A comparative analysis of the efficiency of complex formation of PAMAM-calix-dendrimers and classical PAMAM dendrimers with siRNAs was also performed. The findings may facilitate the creation of novel unique gene delivery systems for cancer nanomedicine development.

## 1. Introduction

Cancer is one of the leading causes of death in the world [1]. Traditional oncology therapies have many limitations, e.g., low specificity, a large number of side effects, limited solubility of drugs in aqueous media, and negative effects on healthy cells and tissues [2,3,4,5,6]. The effectiveness of treatment for some cancers is also challenged by the difficulty of diagnosing and monitoring them [7,8,9]. The above limitations in the available approaches to cancer treatment make it necessary to improve the known and create new ways to control cancer [10,11]. One of the ways to treat cancer is gene therapy [12,13,14], which has already entered the practice of global clinical medicine [15]. The study of the mechanisms of gene activity regulation in recent years has led to the discovery of a new mechanism of gene expression suppression, i.e., RNA interference [16]. RNA interference is the ability of RNA to cause specific degradation of a target matrix RNA whose sequence is complementary to one of the RNA strands [17]. The process of RNA interference is accomplished by short double-stranded RNA molecules called small interfering RNAs (siRNAs). SiRNAs represent a convenient tool that can block the expression of proteins, as the target is determined by the effector sequence. The main barrier to realizing the potential of the RNA interference mechanism as a drug therapy is the delivery of siRNA molecules to the target cells of the affected organ. The delivery process involves transportation to the tissue where the target cells are located, followed by crossing the cell membrane. It is worth noting that naked siRNA is unable to easily cross the cancer cell membrane and deeply penetrate the tumor tissues due to its polyanionic characteristics [18,19,20]. To work efficiently, siRNAs must also be located not only inside the cell of interest but, in particular, in the cytosol where protein synthesis occurs [21,22]. However, the lack of cost-effective, efficient, and low-toxicity carriers of therapeutic nucleic acids remains a major challenge in the clinical application of siRNAs in the gene therapy concept [23].

In recent decades, controlled targeted drug delivery systems have become widespread due to their great potential in solving many problems characteristic of existing dosage forms, e.g., high dosages, non-targeted action, inability to maintain optimal therapeutic concentrations, and low biocompatibility [24]. Such systems show excellent therapeutic effects in the treatment of various cancers and other diseases [25,26,27,28]. The gene delivery vectors known to date can be divided into two large major classes, i.e., viral and non-viral (or synthetic) vectors. Viral vectors are effective but have serious drawbacks such as high immunogenicity and in vivo carcinogenicity [29]. Difficulties associated with the large-scale production of viral carriers result in high prices, making them unaffordable for most patients [30]. Compared to viral systems, synthetic (non-viral) systems are characterized by lower efficiency but greater flexibility and safety. Thus, non-viral vectors are the preferred choice for clinical trials.

The use of macrocyclic compounds (cyclodextrins, (thia)calixarenes, pillararenes, cucurbituriles, etc.) seems to be a promising approach for creating systems for controlled drug delivery [24,31,32,33,34,35,36,37]. The combination of a hydrophobic platform and hydrophilic substituents in the structure of macrocyclic compounds opens up the possibility of designing structures capable of simultaneously binding drugs via hydrophobic and van der Waals interactions and releasing them under the action of external stimuli (pH, temperature, electrochemical or electromagnetic effects, etc.) [24]. In addition, rigid macrocyclic structures have advantages over linear structures in membrane permeability, metabolic stability, and overall pharmacokinetics [38,39]. The presented features of macrocyclic compounds can improve the stability, biocompatibility, drug-loading ability, and tissue permeability of drug delivery systems and increase the efficacy and safety of immunomodulatory therapies. (Thia)calixarene derivatives are some of the most studied macrocycles, but there are few publications to date on the use of these macrocyclic derivatives in the delivery of genetic material (RNA or DNA). Macrocycles containing alcohol, amine, polyamidoamine, tetraalkylammonium, guanidinium, and imidazolium moieties were used for this purpose [40,41,42,43,44,45,46,47]. There are only a few publications in which the binding of (thia)calixarene derivatives to RNA has been studied [48,49,50]. However, there are no examples of the use of such macrocycles for the binding and delivery of therapeutic nucleic acids in the literature. Thus, despite the fact that there are a number of studies devoted to nucleic acid binding by macrocyclic compounds, in particular, (thia)calixarenes, the potential of their use as gene vectors has not yet been fully realized and requires further study.

Dendrimers are widely used as non-viral vectors for gene therapy because of their monodispersity, stability, low viscosity with high molecular weight, and multiple end groups that can be ionized [51,52]. This makes dendrimers able to efficiently bind a large number of genetic material through electrostatic interactions [53,54] and circulate in the body for a long time [55]. The main problem with the use of high-generation dendrimers in clinical practice is their high toxicity and cost of synthesis. Earlier, to solve this problem, our research group proposed the idea of replacing the core and one or more inner “layers” of the dendrimer with macrocyclic compounds close in size, i.e., thiacalixarenes [56]. It was shown that the obtained **PAMAM-calix-dendrimers** can efficiently bind low and high molecular weight substrates to form nanoparticles [45,57,58,59].

In this work, we first proposed the idea of using symmetric **PAMAM-calix-dendrimers** of the first, second, and third generation to bind therapeutic siRNAs and deliver genetic material into cancer cells (Figure 1). The hemo- and cytotoxicity of the obtained macrocyclic dendrimers and their complexes with siRNA were evaluated. In addition, the effect of the thiacalix[4]arene platform on complexation with siRNA compared to classical PAMAM dendrimers was studied. The obtained supramolecular **PAMAM-calix-dendrimer**/siRNA systems can be used to create promising cancer nanovaccines and gene delivery systems.

## 2. Results and Discussion

### 2.1. Design, Synthesis, and Self-Association of PAMAM-Calix-Dendrimers

The design of dendrimeric structures based on macrocycles, including (thia)calixarenes, is a promising direction for the development of modern organic and supramolecular chemistry due to the possibility of overcoming the disadvantages inherent in higher-generation dendrimers, such as high toxicity and synthesis cost. However, only single examples of obtaining first- and second-generation dendrimers with a calix[4]arene core are found in the literature [60,61,62]. Previously, our research group hypothesized the replacement of one or more internal generations of classical PAMAM dendrimers with a thiacalix[4]arene core to increase dendron lability and the volume of dendrimer internal cavities, which could lead to more efficient binding of target substrates. Therefore, the substrate binding efficiency of lower-generation **PAMAM-calix-dendrimers** can be expected to be at the level of classical PAMAM dendrimers of higher generations. The toxicity of PAMAM dendrimers with terminal amino groups is known to increase with their generation and is due to the presence of a large positive charge on the surface of the macromolecule [63,64]. All these points will allow us to avoid the use of toxic and expensive classical PAMAM dendrimers of 4–6 generations and efficiently use **PAMAM-calix-dendrimers** of lower generations for substrate binding and delivery.

Thus, the objects of this study are symmetric **PAMAM-calix-dendrimers** of 1–3 generations with a macrocyclic core in *1*,*3-alternate* conformation (**G1-alt**, **G2-alt**, and **G3-alt**), which were synthesized according to the literature methods in several steps (Figure 2) [45,58,59]. Brief synthetic protocols and ^1^H NMR spectra are summarized in Section 3.1 and the Appendix A). Classical PAMAM dendrimers of similar generations (**PAMAM G1**, **PAMAM G2**, and **PAMAM G3**) were chosen as comparison objects.

To evaluate our hypothesis about the influence of the macrocyclic core on the dimensional characteristics of **PAMAM-calix-dendrimers**, quantum chemical calculations were carried out by the semiempirical PM3 method [65,66] with the use of the Gaussian 16 suite of programs [67]. Quantum chemical calculations for classical PAMAM dendrimers were also performed for comparison. Appendix A shows the optimized structures of the dendrimers. Thus, the shape of **G1-alt** was elongated along one axis, whereas for **G3-alt**, the maximal lengths of molecules for all three directions were close (Appendix A).

For each dendrimer, the molar volume (volume of 1 mole of the dendrimer), molecular volume (volume of 1 molecule of the dendrimer), and radius of gyration (R_g_) were calculated (Table 1 and Appendix A). The calculated values of physical quantities for PAMAM dendrimers were correlated with the literature data [68,69]. Analysis of the computational results showed that the molecular volume in **PAMAM-calix-dendrimer** and classical PAMAM dendrimer series approximately doubles from G1 to G2 and from G2 to G3 (Table 1 and Appendix A). At the same time, the molecular volume of **PAMAM-calix-dendrimers** was larger than the classical PAMAM dendrimers by 15.8–36.5% (G1—34.0%, G2—36.5%, and G3—15.8%). As for R_g_, the calculations predicted the same trend for **PAMAM-calix-dendrimers** as for the classical PAMAM dendrimers. The R_g_ values of **PAMAM-calix-dendrimers** were larger than those generations of the classical PAMAM dendrimers by 1.0–12.4% (G1—1.0%, G2—12.4%, and G3—10.6%). Thus, our hypothesis that the introduction of a macrocyclic core leads to an increase in the molecular size of the target dendrimers is also confirmed by quantum chemical calculations.

The self-organization of molecules plays an important role both in living biological systems and in the creation of synthetic materials [70,71,72,73,74]. As a rule, it is caused by non-covalent interactions, e.g., hydrophobic effects, hydrogen bonds, and electrostatic interactions [75,76,77,78,79]. The study of the self-assembly of molecules in both living and synthetic systems contributes to the development of biomedicine and nanotechnology, as well as the application of the achievements of supramolecular chemistry on living objects. PAMAM dendrimers were previously shown to form polydisperse systems with a wide range of aggregate sizes in water (10–500 nm) [80].

Previously, our research group showed the formation of polydisperse supramolecular systems for **G1-alt**, **G2-alt**, and **G3-alt** in water, PBS, and Tris-HCl buffer at 25 °C [58,59]. Therefore, the next stage of the work was to study the self-association of macrocycles **G1-alt**, **G2-alt**, and **G3-alt** under physiological conditions (PBS, pH = 7.4, 37 °C) in the concentration range of 1–100 µM by dynamic light scattering (DLS) analysis (Table 2). Thus, the size of the **G1-alt** associates (254–289 nm) was independent of the solution concentration (Appendix A), while the polydispersity system index (PDI) values of the studied systems were ≥0.32. Submicron associates were observed for **G2-alt** (Appendix A). Increasing the **G2-alt** concentration from 10 to 100 µM had little effect on the associates’ size. Surprisingly, **G3-alt** formed the most stable particles regardless of the solution concentration (Appendix A). With increasing the solution concentration of **G3-alt** from 1 to 10 µM, there was a decrease in the average hydrodynamic particle diameter (d), while a subsequent tenfold increase in solution concentration had almost no effect on the associates’ size. The PDI values of the systems for **G3-alt** changed nonlinearly with increasing solution concentration (Table 2). Apparently, the formation of the most stable associates of **G3-alt** was due to the presence of a greater number of terminal amino groups and, consequently, the formation of a greater number of hydrogen bonds in PBS.

### 2.2. The Study of Supramolecular Systems Based on PAMAM-Calix-Dendrimers and siRNA

Delivering siRNAs, nucleic acids, and therapeutic agents by classical PAMAM dendrimers has been extensively studied [81,82,83,84]. For these purposes, 6–7 generations of PAMAM dendrimers are considered optimal. The application of differently modified dendrimers of lower generations has also been demonstrated [85,86,87]. However, the known modifications mainly concern the terminal groups. This makes it possible to reduce toxicity, increase the release of target agents in cells, and selectively target dendrimers to specific tissues. Despite the above advantages, surface modifications of classical PAMAM dendrimers always reduce the cationic charge density, and hence, delivery efficiency. In addition, modifications often reduce the capacity of dendrimers’ inner cavities for drug loading. Modifications of the PAMAM dendrimer core are critically rare in the literature and are mainly used in the field of optics [88,89]. There is some work underway to modify the core to encapsulate substances [90,91] and deliver gene material [92,93]. In view of this, the next step was the creation and study of supramolecular systems based on **PAMAM-calix-dendrimers** and siRNAs, and the comparison of such systems with complexes of classical PAMAM dendrimers with siRNAs. Proapoptotic siRNA, i.e., siKRAS and siBCL-2, were used (see Section 3.2). A non-coding non-targeted RNA (ntRNA) was used as a control in experiments where additional toxic effects were undesirable. In the part on determining the binding of dendrimers and siRNA (gel electrophoresis, zeta potential, and DLS), only one siRNA sequence was used since the nucleotides included in the sequence have little effect on complex formation.

#### 2.2.1. Surface Charge of the Dendrimer/siRNA Complexes

The positive surface charge (zeta potential) and the saturation of siRNA complexes are important factors since endocytosis is the main mechanism of penetration in nonspecific interactions with cells [94,95]. Thereby, an approach of complexes to a negatively charged membrane becomes possible only if there is a sufficient cationic charge on the surface of the complexes [96]. This also explains the need for an excess of cationic over anionic charge in the complexes formed. Previous studies of other dendrimer complex formation and internalization have shown that the N:P ratios of 10:1 and above were optimal [97,98].

This rule also appeared to be true given the cellular uptake data for **PAMAM-calix-dendrimers** (see Section 2.3.2). The molecular ratio is related to the N:P ratio according to Equation (2) (see Section 3.4). All studied **PAMAM-calix-dendrimers G1-alt**, **G2-alt**, and **G3-alt** formed stable, charged complexes with siRNAs according to the analysis of complexation by the zeta potential method (Figure 3a). The N:P charge ratio required to saturate the surface charge of the complexes was 7.5:1 for **G2-alt** and **G3-alt** and 10:1 for **G1-alt**. Interestingly, the amount of specific charge required to form neutral complexes decreased as the generation increased.

The nature of the curves indicated different mechanisms of complex formation in the case of **G1-alt** compared to **G2-alt** and **G3-alt**. A change in the zeta potential of the **G1-alt**/siRNA complexes from negative to positive was observed at the N:P 5:1 ratio. In the case of other generations, these values were lower (~3:1 for **G2-alt** and ~2:1 for **G3-alt**). The saturation slope of **G1-alt** was smoother than the saturation curves for **G2-alt** and **G3-alt**, which had a steeper slope but differed in the magnitude of the maximum potential. **PAMAM G1** and **G2** formed complexes with siRNAs with a surface potential close to zero (Figure 3b). The curves for **PAMAM G3** and **G3-alt** were slightly different. In the case of PAMAM dendrimers, only **PAMAM G3** showed a positive electrostatic potential.

#### 2.2.2. Hydrodynamic Diameter of the Dendrimer/siRNA Complexes

The next stage of the work was to establish the size of the dendrimer/siRNA complexes by DLS. The DLS data demonstrated that the hydrodynamic diameters of the **PAMAM-calix-dendrimer**/siRNA complexes were sufficiently large (Figure 4). The hydrodynamic diameter of the **PAMAM-calix-dendrimer**/siBCL-2 complexes decreased with the increasing generation of dendrimers. The average diameter of the **G1-alt**/siBCL-2 complex was ~950 nm. The sizes of the **G2-alt**/siBCL-2 and **G3-alt**/siBCL-2 complexes were ~450 nm and ~300 nm, respectively. A decrease in PDI was observed with the increase in the **PAMAM-calix-dendrimer** generation. Thus, PDI was ~0.6–0.8 for **G1-alt**/siBCL-2 and ~0.3–0.4 for complexes of **G2-alt** and **G3-alt** with siBCL-2. The hydrodynamic diameter of the complexes with PAMAM dendrimers averaged ~750 nm for **PAMAM G3** and ~300–350 nm for **PAMAM G1** and **PAMAM G2**.

Analysis of the DLS data showed no correlations in the hydrodynamic size of the complexed siRNAs with **PAMAM-calix-dendrimers** and classical PAMAM dendrimers. It is well known that, in reality, associates typically possess a significantly smaller size than that which is observed in DLS measurements [99]. This is due to the fact that the method allows for measuring not the complexes themselves but the boundary of the so-called slipping plane, which, in the case of using phosphate-containing solutions, can be a significant fraction of the central complex [100]. It can be assumed that the **PAMAM-calix-dendrimer**/siRNA complexes were close to the optimal size for delivery, which lies in the range of 10–200 nm according to various estimates [101,102]. In addition, some types of endocytosis are also known to be possible for structures up to 1 μm in size [103,104].

#### 2.2.3. Evaluation of the Dendrimer/siRNA Complexes and Their Stability

The visualization of siRNA migration in agarose gel allows for demonstrating the complex formation and estimating the concentration of the compound required for complete binding of the siRNA by **PAMAM-calix-dendrimers** with the formation of stable complexes in all cases (Figure 5 and Appendix A). **G1-alt** demonstrated the highest binding efficiency, i.e., complete siRNA binding was observed at a charge ratio of 2.5:1. Complete binding was reliably observed at values of 7.5:1 and above for **G2-alt** and **G3-alt**. The addition of heparin to the formed complexes contributed to the complete release of siRNAs. This suggests that the interaction between dendrimers and siRNAs is mainly electrostatic (Appendix A). The binding of siRNAs to PAMAM dendrimers was studied to compare the effects of **PAMAM-calix-dendrimers**. PAMAM dendrimers bound genetic material much weaker than all **PAMAM-calix-dendrimers** except for **G3-alt** (Figure 5). No binding was observed for **PAMAM G2**, which might indicate a weak electrostatic interaction between siRNA and the PAMAM dendrimer.

A number of experiments were also carried out to elucidate the stability characteristics. Two N:P ratios were chosen (10:1 and 15:1) for this purpose. The **PAMAM-calix-dendrimer**/siRNA complexes retained their stability (Appendix A) for at least five days. **PAMAM-calix-dendrimers** also showed good protective properties in the presence of RNases (Appendix A). The complexes were treated with a mixture of nucleases, after which RNA was released from the complexes with heparin and stained with ethidium bromide. FAM-labeled RNA was not used in the experiment because the fluorescent label itself, in combination with the dithymine modification, prevents recognition and degradation by RNases.

In the presence of fetal bovine serum, no noticeable degradation of the **PAMAM-calix-dendrimer**/siRNA complexes was observed after either 3 or 24 h of incubation (Appendix A). Moreover, the stability of the **PAMAM-calix-dendrimer**/siRNA complexes was maintained both in the case of serum at the corresponding concentration of the complete medium (10%) and in the presence of an excess amount of serum (50%). In all cases, in the presence of FBS, a barely perceptible fluorescent trace was observed on the gel, which may indicate a slight displacement of siRNA from the complexes and its binding to serum proteins.

Thus, it was shown that **PAMAM-calix-dendrimers** formed stable complexes for all the generations studied. The highest siRNA binding efficiency was observed for **G1-alt**. Thus, complete siRNA binding was observed at the N:P ratio of 2.5:1 for **G1-alt** and 7.5:1 for **G2-alt** and **G3-alt**. A different nature of the interaction was demonstrated for **G1-alt**, which indicated the contribution of the thiacalixarene core to the mechanism of complex formation. All **PAMAM-calix-dendrimers** had superior complexing properties compared to PAMAM dendrimers of the corresponding generations. Good temporal stability was established for all **PAMAM-calix-dendrimer**/siRNAs complexes for at least five days. In addition, the complexes retained stability and protected siRNAs from degradation in the presence of RNases and serum proteins.

### 2.3. Biological Activity of Dendrimer/siRNA Complexes

#### 2.3.1. Toxic Effects of PAMAM-Calix-Dendrimers and PAMAM Dendrimers on Human Peripheral Blood Cells

Since **PAMAM-calix-dendrimers** had positively charged surfaces under physiological conditions, it was necessary to evaluate their toxic effects. These compounds could bind to glycolipids and glycoproteins of erythrocyte membranes because of their positive charge, causing thinning and damage to the membrane and the formation of pores [105,106]. In view of this, toxicity studies were carried out on human red blood cells (RBCs) and peripheral blood mononuclear cells (PBMCs). The hemolytic activity of **PAMAM-calix-dendrimers** was assessed by the level of free hemoglobin released as a result of RBC damage. The concentrations used (20, 10, and 5 μM for **G1-alt**, **G2-alt**, and **G3-alt**, respectively) were chosen based on the N:P ratios in the siRNA complexes, which correlated with other experiments.

**PAMAM-calix-dendrimers** showed generation-dependent toxicity (Figure 6) for RBCs. **G1-alt** had the greatest toxic effects after 3 h of incubation, while **G2-alt** and **G3-alt** were less toxic. The pronounced toxicity of **G1-alt** already after 3 h indicated pronounced primary damage in the erythrocyte membrane and the release of hemoglobin into the environment. Effects observed after 24 h of incubation are usually associated with delayed effects occurring in erythrocytes. These effects may be associated with lipid peroxidation and interaction with cytoplasmic proteins [107,108]. The level of hemolysis changed significantly in the case of **G1-alt** and **G2-alt** after 24 h of incubation, while **G3-alt** retained a low level of toxicity.

Studies of **PAMAM-calix-dendrimers** on PBMCs showed comparable levels of toxicity. However, the differences between the generations became more pronounced with increasing concentration. The dose–response curves plateaued at high concentrations. The viability plateau was higher for the higher-generation dendrimers (Figure 7). **PAMAM-calix-dendrimers** at high concentrations showed higher toxicity on PBMCs compared to classical PAMAM dendrimers. However, **PAMAM-calix-dendrimers** are low-toxic at the concentration required for efficient RNA binding. The IC_80_ at PBMC for all three dendrimers was approximately 5 μM, while the concentrations used for siRNA delivery did not exceed 5.3, 2.6, and 1.3 μM in the case of **G1-alt**, **G2-alt**, and **G3-alt**, respectively.

Similar studies have been reported in the literature for classical PAMAM dendrimers [108,109,110]. Nevertheless, these experiments were carried out under the selected experimental conditions to ensure the accuracy of the comparison. G1–G3 PAMAM dendrimers had low toxicity to blood cells, as described in the literature and confirmed in this work. No significant cytotoxic effects of PAMAM dendrimers on RBCs were detected at appropriate concentrations. PAMAM dendrimers also showed moderate toxicity to PBMCs at high concentrations (Appendix A). It is known that the in vitro and in vivo toxicity of classical PAMAM dendrimers typically increases with increasing generation, especially in the case of large (G4–G6) generations [111,112]. A notable characteristic of **PAMAM-calix-dendrimers** is the unusual decrease in toxicity with increasing compound generation. Such results can be explained by the change in the hydrophilic–lipophilic balance of **PAMAM-calix-dendrimer** molecules, as well as by the lower density of positive surface charge compared to PAMAM dendrimers.

#### 2.3.2. Cellular Uptake of the Dendrimer/siRNA Complexes by HeLa Cells

Flow cytometry data showed that **PAMAM-calix-dendrimers** were internalized to some extent into HeLa cells (Figure 8). In contrast, control measurements of **PAMAM G1** and **PAMAM G2** showed no significant level of cellular uptake of the complexes (Appendix A). The complexes based on **PAMAM G3** reached ~60% of cellular uptake after 24 h of incubation.

A high level of internalization was observed in all cases for the complexes based on **G1-alt**, reaching 80% after one day of incubation at the N:P ratios of 10:1 and 15:1. **G2-alt** showed the lowest level of internalization. The median fluorescence intensity was also the lowest among the other compounds, indicating a low level of accumulation of the complexes in cells. A high level of internalization (86.1 ± 4.5%) with the N:P ratio of 15:1 was found for **G3-alt**. The low uptake of **G2-alt** may be due to a lower level of siRNA binding, as indicated by the results of gel electrophoresis and zeta potential. This in turn may be due to the spatial structure of binding. **G2-alt** has a structure that is not close enough to classical dendrimers and does not have high branch lability compared to **G3-alt**, but at the same time, it already more tightly closes the hydrophobic cavities into which siRNA could be embedded. The complexes based on **G1-alt** accumulated in large vesicles in the cell after 24 h of incubation (Appendix A) according to fluorescence microscopy data. At the same time, for **G2-alt** and **G3-alt**, no inclusions of complexes were detected in the images.

Unlike classical PAMAM dendrimers, **PAMAM-calix-dendrimers** do not exhibit similar patterns associated with their physicochemical properties. This is primarily due to the relatively large size of the thiacalixarene core, which is comparable in linear dimensions to the branches of first-generation dendrimers. This leads to the fact that the nature of uptake, which usually depends on the generation of classical dendrimers, in our case also strongly depends on the core, which makes it problematic to identify relationships only on three generations of compounds. For example, a low level of cellular uptake would be expected for the **G1-alt**/siRNA complexes at the N:P ratio of 5:1, since the zeta potential of these complexes is barely above zero at this concentration and the hydrodynamic diameter reaches 1 μm. However, even at these ratios, the **G1-alt**/siRNA complexes have the ability to penetrate into cells at a fairly high level. This appears to be due to a mechanism of entry that appears to be different in the case of **G1-alt** than other compounds and results in membrane damage. At the same time, the less toxic **G2-alt** is poorly internalized by cells because the influence of the thiacalixarene core is less significant, and its surface potential is apparently insufficient for cell penetration.

#### 2.3.3. Cytotoxic Effect of Dendrimers and Their Complexes with siRNAs on HeLa Cells

In the course of this study, the cytotoxic effects of **PAMAM-calix-dendrimers** and their complexes with siRNAs on HeLa cells were observed. The cells were incubated in the presence of **PAMAM-calix-dendrimers** for 24 and 72 h (Figure 9). Cytotoxic activity on HeLa cells depended on the generation of **PAMAM-calix-dendrimers**. Thus, the **PAMAM-calix-dendrimers** had the same cytotoxic pattern for both blood and HeLa cells. IC_50_ was ~20 μM after 72 h of incubation in the case of **G1-alt**. The corresponding values of IC_50_ were 58 and 87 μM for **G2-alt** and **G3-alt**, respectively. Control measurements of PAMAM dendrimers of all three generations showed that the toxic effects were rather insignificant (Appendix A). HeLa viability reached a minimum at a plateau of ~70–90% depending on the generation of the dendrimers at their high concentrations. **G2-alt** and **G3-alt** retained low toxicity over a fairly wide range of concentrations. It should be noted that the higher the generation of dendrons, the greater the interval over which the cytotoxicity curves of classical PAMAM dendrimers and **PAMAM-calix-dendrimers** coincide with an accuracy of measurement error. Thus, for **G1-alt**, the cytotoxic effect curves coincide up to concentrations of 15 μM; for **G2-alt** and **G3-alt**, the curves coincide up to concentrations of 50 μM and 80 μM, respectively (Figure 9 and Appendix A).

Finally, the ability of the complexes with targeted siRNA to initiate additional cell death was tested to indirectly confirm the release of siRNA from the complexes and their gene silencing. This test is a quick and easy way to make a preliminary assessment of the efficacy of newly synthesized compounds. The complexes with control RNA (ntRNA) and a series of pro-apoptotic siRNAs, designed to block anti-apoptotic proteins in tumor cells, were used in the experiment. Based on our previous results, **PAMAM-calix-dendrimer** concentrations corresponding to the N:P ratio of 10:1 were selected. The siRNA concentration was 100 nM. **PAMAM-calix-dendrimers** did not induce toxic effects on HeLa cells at appropriate concentrations. The **PAMAM-calix-dendrimer**/ntRNA complexes also did not cause an increase in cell death compared to untreated control cells. A significant difference in viability between cells treated with target siRNA complexes and cells treated with control RNA complexes was observed only for siKRAS in the complex based on **G3-alt** (Figure 10). This difference indicated a low release of siRNA from the **G3-alt**/siKRAS complexes. We performed screening studies of PAMAM G3, which confirmed the literature data that classical unmodified compounds do not release siRNA in cells [113]. Thus, the obtained **PAMAM-calix-dendrimer**/siRNA complexes can be used for the slow release of siRNA and the creation of prolonged action systems.

Thus, the cytotoxic activity of **PAMAM-calix-dendrimers** on HeLa cells was also dependent on dendron generation. **G1-alt** demonstrated the highest toxicity, while the cytotoxicity of **G2-alt** and **G3-alt** was significantly lower. The cytotoxic effect of **PAMAM-calix-dendrimers** correlated with that of PAMAM dendrimers prior to the drop in survival on dose–response curves. On the other hand, PAMAM dendrimers had no or negligible toxic effects with a decrease in cell viability to a level of about 70%. However, **PAMAM-calix-dendrimers** did not cause toxic effects at concentrations required for siRNA delivery.

The complexes based on all the studied generations of **PAMAM-calix-dendrimers** showed good internalization ability, with the greatest effect observed for **G1-alt**. The level of their accumulation was extremely high. However, fluorescence microscopy showed that complexes based on **G1-alt** formed bright inclusions in the cells and did not decompose after one day of incubation, indicating a low ability of siRNA to leave these complexes. In addition, some of the complexes are located outside the cells or are associated with cell debris, which suggests that not all **G1-alt** complexes penetrate into cells. These problems can be associated both with a high packing density due to the high lability of the dendrimer branches and with the size of the complexes, which led to a disruption of the lysosome formation process. Complexes based on **G2-alt** accumulated less in cells than in other generations. **G2-alt** could not accumulate sufficiently in the cell despite being close in all respects to **G3-alt**, which not only accumulated well in cells but also managed to release some of the genetic material. Considering the similar size of the complexes based on **G2-alt** and **G3-alt**, but significantly different surface potential indicators, it can be assumed that for this type of nanoparticles, it is also necessary to overcome a certain potential barrier in order for the complexes to effectively penetrate into the cells under study.

## 3. Materials and Methods

### 3.1. Synthesis

First, second and third generation PAMAM dendrimers (**PAMAM G1**, **PAMAM G2,** and **PAMAM G3**) with ethylenediamine core (Aldrich) were used as received without additional purification.

**PAMAM-calix-dendrimers G1-alt**, **G2-alt** and **G3-alt** were synthesized by the previously described protocols [45,58,59]. Brief synthetic protocols are summarized below (Figure 1).

For half-generations (ester derivatives):

The solution (10% by the weight) of the precursor amino derivative in methanol was added dropwise to an ice-cooled methyl acrylate (4-fold excess per amino group) dissolved in equal amount of methanol. The reaction mixture was stirred at room temperature for 12–48 h depending on generation (Figure 1a,c,e). Afterward, the remaining methyl acrylate and the solvent were removed on a rotary evaporator. The residue was dried under reduced pressure.

For full-generations (amino derivatives):

The solution (10% by the weight) of the precursor ester derivative in methanol was added dropwise to an ice-cooled (0 °C) solution of ethylenediamine (20-fold excess per ester group) in equal amount of methanol. The reaction mixture was stirred at room temperature for 70–110 h depending on generation (Figure 1b,d,f). The solvent then was removed on a rotary evaporator, and the excess of ethylenediamine was removed by azeotropic distillation with the mixture methanol: toluene (1:9). Then, the remaining toluene was removed by distillation with methanol. The residue was dried under reduced pressure.

[**G1-alt]** 5,11,17,23-tetra-*tert*-butyl-25,26,27,28-tetrakis[*N*-(6-(*N*,*N*-di(*N*-(2-aminoethyl)carbamoylethyl)amino)hexyl)carbamoylmethoxy]-2,8,14,20-tetrathiacalix[4]arene in *1,3-alternate* conformation. White solid foam. Yield 99%.

^1^H NMR (CD_3_OD, δ, ppm, *J*/Hz): 1.28 (s, 36H, (CH_3_)_3_C), 1.30–1.39 (m, 16H, C(O)NHCH_2_CH_2_CH_2_CH_2_), 1.42–1.61 (m, 16H, CH_2_CH_2_NH_2_, C(O)NHCH_2_CH_2_CH_2_CH_2_), 2.37 (t, 16H, NCH_2_CH_2_C(O), ^3^*J*_HH_ = 6.9), 2.46 (br.t, 8H, CH_2_CH_2_N), 2.64–2.82 (m, 32H, NHCH_2_CH_2_NH_2_, NCH_2_CH_2_C(O)), 3.12–3.29 (m, 24H, NHCH_2_), 4.15 (br.s, 8H, OCH_2_C(O)), 7.59 (s, 8H, ArH).

[**G2-alt**] 5,11,17,23-tetra-*tert*-butyl-25,26,27,28-tetrakis[*N*-(6-(*N*,*N*-di(N-(2-(*N*,*N*-di(*N*-(2-aminoethyl)carbamoylethyl)amino)ethyl)carbamoylethyl)amino)hexyl)carbamoylmethoxy]-2,8,14,20-tetrathiacalix[4]arene in *1,3-alternate* conformation. White solid foam. Yield 88%.

^1^H NMR (CD_3_OD, δ, ppm, *J*/Hz): 1.28 (s, 36H, (CH_3_)_3_C), 1.31–1.37 (m, 16H, C(O)NHCH_2_CH_2_CH_2_CH_2_), 1.45–1.60 (m, 16H, CH_2_CH_2_CH_2_N, C(O)NHCH_2_CH_2_CH_2_CH_2_), 2.33–2.42 (m, 48H, NCH_2_CH_2_C(O)), 2.44–2.52 (m, 8H, CH_2_CH_2_CH_2_N), 2.57 (t, 16H, NHCH_2_CH_2_N, ^3^*J*_HH_ = 6.8), 2.72 (t, 32H, NHCH_2_CH_2_NH_2_, ^3^*J*_HH_ = 6.3), 2.75–2.84 (m, 48H, NCH_2_CH_2_C(O)), 3.17–3.29 (m, 54H, NHCH_2_), 4.12 (br.s, 8H, OCH_2_C(O)), 7.60 (s, 8H, ArH).

[**G3-alt**] 5,11,17,23-tetra-*tert*-butyl-25,26,27,28-tetrakis[*N*-(6-(*N*,*N*-di(*N*-(2-(*N*,*N*-di(*N*-(2-(*N*,*N*-di(*N*-(2-aminoethyl)carbamoylethyl)amino)ethyl)carbamoylethyl)amino)ethyl)carbamoylethyl)amino)hexyl)carbamoylmethoxy]-2,8,14,20-tetrathiacalix[4]arene in *1,3-alternate* conformation. White solid foam. Yield 84%.

^1^H NMR (CD_3_OD, δ, ppm): 1.28 (s, 36H, (CH_3_)_3_C), 1.29–1.38 (m, 16H, C(O)NHCH_2_CH_2_CH_2_CH_2_), 1.44–1.64 (m, 16H, CH_2_CH_2_CH_2_N, C(O)NHCH_2_CH_2_CH_2_), 2.32–2.42 (m, 112H, NCH_2_CH_2_C(O)), 2.44–2.52 (m, 8H, CH_2_CH_2_CH_2_N), 2.53–2.63 (m, 48H, NHCH_2_CH_2_N), 2.69–2.86 (m, 176H, NHCH_2_CH_2_NH_2_, NCH_2_CH_2_C(O)), 3.17–3.30 (m, 120H, NHCH_2_), 4.00–4.21 (br.s, 8H, OCH_2_C(O)), 7.60 (s, 8H, ArH).

### 3.2. Small Interfering RNAs

Proapoptotic siRNAs (siKRAS and siBCL-2) were used (Table 3). As a control, and for experiments where additional toxic effects are undesirable, a control non-coding non-targeted RNA (ntRNA) was used. The terminal areas were modified with ditimine to increase RNA stability and provide partial protection against natural RNases.

### 3.3. Molecular Modeling

The structures of **PAMAM-calix-dendrimers** have been optimized by semiempirical PM3 method [65,66] with the use of Gaussian 16 suite of programs [67]. Radius of gyration (*R_g_*) has been calculated by Equation (1).
(1)$$R_{g}=\sqrt{\frac{\sum_{i=1}^{n}m_{i}\cdot {\left(r_{i}-R\right)}^{2}}{M}}$$
where *R* is the center-of-mass of the dendrimer, *m_i_* is the mass of the *i*^th^ atom, and *M* is the total mass of the dendrimer.

### 3.4. Dendrimer/siRNA Complex Preparation

In all studies of complex formation and the effect of the resulting complexes on biological objects in vitro, complexes of **PAMAM-calix-dendrimers** with siRNA were prepared as follows. A stock solution of **PAMAM-calix-dendrimers** in deionized water and a stock solution of siRNA in storage buffer (60 mM KCl, 6 mM HEPES-pH 7.5, 0.2 mM MgCl_2_) were mixed in PBS in the corresponding N:P ratio. Before further measurements or introduction to biological objects, the samples were incubated for 15 min at room temperature (22 °C). The molecular ratio is related to the N:P ratio according to Equation (2).
(2)$$\;N:P\;\mathrm{ratio}=\frac{\left[\mathrm{dendrimer}\right]}{\left[\mathrm{siRNA}\right]}\times \frac{N}{P}$$
where [dendrimer] and [siRNA] are molar concentrations of dendrimer and siRNA, respectively; N is the number of terminal amines of **PAMAM-calix-dendrimer** (8, 16, and 32 for **G1-alt**, **G2-alt**, and **G3-alt** respectively); P is the number of deprotonated phosphates in siRNA in PBS (42 for all cases). Table 4 shows the molar concentrations of dendrimer per 1 μM siRNA.

### 3.5. Dynamic Light Scattering (DLS)

#### 3.5.1. Self-Assembly of PAMAM-Calix-Dendrimers

The distribution of particles by number, volume, and intensity, the polydispersity index were determined by DLS on a Zetasizer Nano ZS instrument (Malvern Instruments, Worcestershire, UK) in quartz cuvettes. The instrument is equipped with the 4 mW He-Ne laser (633 nm). Measurements were performed at a detection angle of 173°. The error in determining the particle size is less than 2%. The results were processed by the DTS program (Dispersion Technology Software 4.20). The self-assembling data were recorded at 37 °C in PBS (pH = 7.4) 1 h after preparation. In the course of the experiment, the concentrations of **PAMAM-calix-dendrimers** varied within 1–100 μM. The particle sizes were measured after 1 h mixing. Three independent experiments were carried out for each series. Measurements were determined after 24 and 178 h three times to evaluate kinetic stability.

#### 3.5.2. Aggregation of the Dendrimers with siRNA

Hydrodynamic diameter and size distribution of the complexes at different charge ratios were measured by DLS using Malvern Zeta Sizer-Nano-ZS photon correlation spectrometer (Malvern Instruments Ltd., Worcestershire, UK) in DTS0012 plastic cuvettes. The following parameters were used in the device settings: refractive index 1.33, detection angle 90°, and red laser with a wavelength of 633 nm. Measurements were carried out at a concentration of 0.5 μM siRNA (siBCL-2) in 1X PBS (pH = 7.4). Increasing concentrations of the dendrimers were measured with fixed concentration of siRNA. The samples were incubated for 10 min. Hydrodynamic size of the complexes was measured on average over 12 cycles of 4 independent measurements at 37 °C. Malvern software (version 7.11) was used to analyze the data. The polydispersity index (PDI) was used as an additional parameter.

### 3.6. Zeta-Potential of the Dendrimer/siRNA Complexes

The surface potential of the complexes based on the dendrimers and siRNA (siBCL-2) was measured by phase analysis light scattering (PALS) with a Malvern Instruments Zetasizer Nano-ZS (Malvern Instruments Ltd., Worcestershire, UK). DTS1061 plastic capillary cells (Malvern) were used to measure the electrophoretic mobility of the samples in applied electric field. Zeta-potential of siRNA was measured at concentration of 0.5 μM. The remaining samples measured contained the same amount of siRNA and different concentrations of the dendrimers under investigation to plot the saturation curve. The complexes were incubated for 10 min and then measured at 37 °C. On average, 5–7 independent measurements of 15 replicates each were made for each sample. The Helmholtz-Smoluchowski equation was used to calculate zeta potential values in the Malvern software.

### 3.7. One-Dimensional Agarose Gel Electrophoresis

One-dimensional agarose gel electrophoresis was used to determine the degree of genetic material binding to the dendrimers. A mixture of FAM-labeled siRNAs (siBCL-2, 1.5 μM) and **PAMAM-calix-dendrimers** at various charge ratios was prepared in PBS, pH 7.4, after which it was incubated for 15 min at room temperature and the samples were added to the wells, preliminarily with 30% glycerol as loading buffer in 1:5 ratio. To determine the temporal stability of the complexes, concentrations of **PAMAM-calix-dendrimers** were chosen at which complete siRNA binding was observed on the basis of previous experiments. Samples were prepared in a similar manner and incubated for various time, up to 5 days. To determine the stability of the complexes under the action of RNases, complexes with unlabeled RNA were subjected to treatment with an RNase A/T1 Mix (RNase A/T1 Mix, Fisher Scientific UK Ltd., Loughborough, UK) according to the manufacturer’s standard protocol. The complexes were treated with heparin and run on the gel using standard electrophoresis parameters. After completion of electrophoresis, the gel was stained with ethidium bromide (MilliporeSigma, Burlington, MA, USA) to visualize siRNA. To determine the stability of the complexes in the serum medium, 10% or 50% solution of fetal bovine serum (FBS, Capricorn Scientific GmbH, Ebsdorfergrund, Germany) was added to the complexes prepared according to the procedure described above. The complexes were incubated for 3 or 24 h in serum.

Electrophoresis was performed in 3% agarose gel for 30 min at 40 V. Tris-acetate buffer (TAE) was used as the electrophoretic buffer. The gel was then visualized under ultraviolet light using digital photography. Imaging of gel electrophoresis results was performed on the VersaDoc™ (Bio-Rad Laboratories, Inc., Hercules, CA, USA). The signal was recorded using a CCD matrix.

The complexes formation was evaluated by slowing down the migration of FAM-labeled siRNA.

### 3.8. Hemolysis

Spectroscopic measurements were made to assess the effect of the dendrimers on human erythrocyte hemolysis. This experiment used whole blood from donors. The blood was centrifuged at 3000× *g* for 5 min at 4 °C to precipitate red blood cells (RBCs), then plasma and peripheral blood mononuclear cells (PBMCs) were removed. RBCs were washed 2 times with cold PBS (4 °C) until transparent, centrifuged at 3000× *g* for 5 min at 4 °C.

Samples were prepared in PBS; RBC was diluted to achieve 2% hematocrit. 2% hematocrit was prepared in distilled water (hypotonic solution which leads to full hemolysis) and used as a positive control. Solutions containing the dendrimers were then added to the samples and incubated for the required time. After the incubation, the samples were centrifuged at 1000× *g* for 10 min at room temperature, positive controls were centrifuged at 5000× *g* for 5 min. The samples were then measured at a fixed wavelength of 540 nm. 

### 3.9. Peripheral Blood Mononuclear Cell (PBMC) Inhibition

PBMCs were seeded into 96-well black plates at a density of 1 × 10^5^ cells per well (1 × 10^6^ cells per mL). The cells were treated with **PAMAM-calix-dendrimers** at various concentrations for 72 h. After the incubation, resazurin was added to the culture medium to a final concentration of 10 μg/mL. The plates were incubated at 37 °C in the dark to allow the conversion of resazurin to resorufin. Fluorescence of metabolized resazurin in the cell suspension was measured after 90 min at 530 nm excitation and 590 nm emission using a Wallac 1420 Multilabel Counter (Wallac Oy PerkinElmer, Turku, Finland). Cell viability was calculated relative to the control (untreated).

### 3.10. Cell Line Cultures

Cytotoxicity and cellular uptake experiments were carried out on HeLa (human cervical adenocarcinoma). Cells were grown in DMEM (glucose content–4.5 g/L, L-lutamine–4 mM; Life Technologies, Paisley, UK), supplemented by inactivated fetal bovine serum (FBS; 10% in volume; Life Technologies, Paisley, UK) and antibiotics streptomycin 100 mg/mL and penicillin 100 U/mL (MilliporeSigma, Burlington, MA, USA). Cells were cultivated in cultural flasks with 75 cm^2^ surface area and standard adhesion at 37 °C in a humidified air atmosphere with 5% CO_2_ content and were passaged 2–3 times a week when 80% confluence reached.

### 3.11. Cytotoxicity Studies

The cell lines were seeded at 1 × 10^5^ cells/mL in 100 μL of suitable culture medium on 96-well plates (1 × 10^4^ cells/well). They were preincubated for 24 h and a further 24 h or 72 h after adding complexes. Complexes for treatment were incubated for 15 min in PBS at concentrations required to deliver 100 nM siRNA. Cytotoxicity was evaluated by the MTT-test (Carl Roth, Karlsruhe, Germany) for HeLa cells. Absorbance measurements used a Wallac 1420 Multilabel Counter (Wallac Oy PerkinElmer, Turku, Finland) at λ_abs_ = 570 nm, using a reference wavelength of 630 nm for MTT. Cell viability (relative units—r.u.) was calculated relative to the control (untreated).

### 3.12. Cellular Uptake

The cell lines were seeded at 2 × 10^5^ cells/mL in 500 μL of suitable culture medium on 24-well plates (1 × 10^5^ cells/well) and preincubated for 24 h at 37 °C in humidified air atmosphere with 5% CO2 before treatment. Complexes of FAM-labeled siRNA and **PAMAM-calix-dendrimers** were added to the cells. Complexes for treatment were incubated for 15 min in PBS at concentrations required to deliver 100 nM siRNA-FAM. Cells were incubated for 3 or 24 h and washed in PBS according to the standard procedure.

Samples were analyzed by flow cytometry (CytoFLEX, Beckman Coulter, Indianapolis, IN, USA): 25,000 events with the exclusion of aggregated (SSC-H/SSC-A gate) cells.

## 4. Conclusions

In this work, non-viral delivery systems for regulatory siRNAs into cancer cells have developed based on polyamidoamine (PAMAM) dendrimers with a thiacalixarene core (**PAMAM-calix-dendrimers**) in *1,3-alternate* stereoisomeric form. **PAMAM-calix-dendrimers** of the first, second, and third generation superiorly bound siRNA and formed positively charged submicron size (300–950 nm) complexes. Complete siRNA binding was observed at the N:P ratio of 2.5:1 for **G1-alt** and 7.5:1 for **G2-alt** and **G3-alt**. The obtained **PAMAM-calix-dendrimer**/siRNA complexes successfully internalized into cancer cells and partially suppressed their activity in the case of **G3-alt**. Good temporal stability was established for all **PAMAM-calix-dendrimer** complexes with siRNA. In addition, the complexes retained stability and protected siRNAs from degradation in the presence of RNases and serum proteins. **PAMAM-calix-dendrimers** showed the unusual decrease in hemo- and cytotoxicity with increasing generation and did not cause significant toxic effects on blood cells at concentrations required for siRNA binding and delivery. The IC_80_ at PBMC for all three compounds was approximately 5 μM, while the concentrations used for siRNA delivery did not exceed 5.3, 2.6, and 1.3 μM in the case of **G1-alt**, **G2-alt**, and **G3-alt**, respectively. The existing problem of **PAMAM-calix-dendrimer** toxicity at high concentrations can be solved by modifying the terminal groups in our future studies. The effect of the thiacalix[4]arene platform on complexation with siRNA compared to classical PAMAM dendrimers was studied. A fundamental feature of **PAMAM-calix-dendrimers** was the high efficiency of siRNA binding compared to classical PAMAM dendrimers of the same generations. The obtained supramolecular **PAMAM-calix-dendrimer**/siRNA systems can be used to create promising cancer nanomedicines and gene delivery systems.

## Data Availability

The data presented in this study are available in the Appendix A.

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
