# Peer review of "Non-Viral Systems Based on PAMAM-Calix-Dendrimers for Regulatory siRNA Delivery into Cancer Cells"

_ijms, 2024, doi:10.3390/ijms252312614_

Round 1
Reviewer 1 Report
Comments and Suggestions for Authors
The manuscript submitted by Padnya et al. investigates the synthesis and characterization of a series of specific dendrimers and their complex formation with the delivery of siRNA. Moreover, the authors have compared these PAMAMA-calix-dendrimers with more classical ones in order to demonstrate their better efficiency. The manuscript is clear, well written and the conclusions are supported by the results. However, some corrections are needed in order to increase the overall quality of the paper:
1. line 237: zeta potential for G1-alt is almost equal to the G3-alt value, within the experimental error limits.
2. the stability of the complexes must be also determined by DLS as a function of time and temperature.
3. complete the sentence from line 306: "the highest siRNA binding efficiency observed for G1-alt."
4. line 346-348: indicate the concentration from which the samples are cytotoxic.
5. more specific results must be provided in the conclusion section.
Author Response
Reviewer #1:
The manuscript submitted by Padnya et al. investigates the synthesis and characterization of a series of specific dendrimers and their complex formation with the delivery of siRNA. Moreover, the authors have compared these PAMAMA-calix-dendrimers with more classical ones in order to demonstrate their better efficiency. The manuscript is clear, well written and the conclusions are supported by the results.
Response:
Dear Reviewer! Thank you very much for carefully reading and reviewing our paper! We are very pleased that you have given it such a high rating.
However, some corrections are needed in order to increase the overall quality of the paper:
- line 237: zeta potential for G1-alt is almost equal to the G3-alt value, within the experimental error limits.
Response:
We meant that the change from negative to positive zeta potential values of PAMAM-calix-dendrimers/siRNA complexes is observed at different N:P ratios. This part has been rewritten for better understanding:
"A change in the zeta potential of the G1-alt/siRNA complexes from negative to positive is observed at the N:P 5:1 ratio. In the case of other generations, these values were lower (~3:1 for G2-alt and ~2:1 for G3-alt)."
- the stability of the complexes must be also determined by DLS as a function of time and temperature.
Response:
Temporal stability was determined using agarose gel electrophoresis. After five days, the samples were visually transparent and did not form visible aggregates, so the DLS was not measured after that time. In addition, during the gel electrophoresis experiments, multiple screening studies were carried out (not presented in the manuscript due to low informativeness) under various conditions, including in complete growth medium at 37 °C. In these cases, too, no turbidity or formation of visible aggregates was observed. We agree with the reviewer that such studies are necessary for a more detailed study of stability. We will definitely perform similar DLS studies in future work with modified PAMAM-calix-dendrimers.
- complete the sentence from line 306: "the highest siRNA binding efficiency observed for G1-alt."
Response:
This part has been rewritten for better understanding. The following fragment was added to the manuscript text:
"Thus, complete siRNA binding is observed at the N:P ratio of 2.5:1 for G1-alt and 7.5:1 for G2-alt and G3-alt."
- line 346-348: indicate the concentration from which the samples are cytotoxic.
Response:
The necessary information has been added to the manuscript text:
"The IC80 at PBMC for all three dendrimers was approximately 5 μM, while the concentrations used for siRNA delivery did not exceed 5.3, 2.6, and 1.3 μM in the case of G1-alt, G2-alt, and G3-alt, respectively."
- more specific results must be provided in the conclusion section.
The conclusion section has been rewritten.
Reviewer 2 Report
Comments and Suggestions for Authors
Very relevant paper highlighting dendrimer-based delivery platforms for nucleic acids into cancer cells. The paper is nicely written, the methodology appropriate and the results seem quite clear. The main flaw is the lack of physico-chemical properties-activity relationships, this point should be improved (see comment below):
1) The authors mention that non specific interactions are essential for endocytosis, however they don't specifically mention them. They should clarify whether they mean electrostatic interactions between the cell lipid membrane and the delivery platforms, the stronger the charge, the more the material penetrates into the bilayer. If this is the case, they should refer to a recent fundamental study that has proven this via nanoscale tools such as neutron reflectivity: M.E. Villanueva et al., Resolving the interactions hydrophilic CdTe quantum dots and positively charged membranes at the nanoscale, Journal of Colloid and Interface Science 677, 620 (2025)
2) They mention at some point (line 266) the optimal size for delivery but they do not specify the size nor the reports in the literature indicating this size.
3) What is the zeta potential of the cells under study (RBCs and hella cells)?
4) The authors should include more detailed information on the correlation between physico-chemical properties of the delivery platforms (size, zeta potential) and the performance on uptake.
Author Response
Reviewer #2:
Very relevant paper highlighting dendrimer-based delivery platforms for nucleic acids into cancer cells. The paper is nicely written, the methodology appropriate and the results seem quite clear.
Response:
Dear Reviewer! Thank you very much for carefully reading and reviewing our paper! We are very pleased that you have given it such a high rating.
The main flaw is the lack of physico-chemical properties-activity relationships, this point should be improved (see comment below):
1) The authors mention that non specific interactions are essential for endocytosis, however they don't specifically mention them. They should clarify whether they mean electrostatic interactions between the cell lipid membrane and the delivery platforms, the stronger the charge, the more the material penetrates into the bilayer. If this is the case, they should refer to a recent fundamental study that has proven this via nanoscale tools such as neutron reflectivity: M.E. Villanueva et al., Resolving the interactions hydrophilic CdTe quantum dots and positively charged membranes at the nanoscale, Journal of Colloid and Interface Science 677, 620 (2025)
Response:
We agree that this part requires reference to experimentally confirmed mechanisms. The necessary references have been added to the manuscript text.
2) They mention at some point (line 266) the optimal size for delivery but they do not specify the size nor the reports in the literature indicating this size.
Response:
Thanks for the note. The preferred size range of nanocomplexes with literature references was added in the manuscript text.
"It can be assumed that the PAMAM-calix-dendrimer/siRNA complexes were close to the optimal size for delivery, which lies in the range of 10–200 nm according to various estimates [101, 102]. In addition, some types of endocytosis are also known to be possible for structures up to 1 μm in size [103, 104]."
3) What is the zeta potential of the cells under study (RBCs and hella cells)?
Response:
According to the data [Bondar O. V. et al. Monitoring of the zeta potential of human cells upon reduction in their viability and interaction with polymers // Acta Naturae, 2012, 4(1), 78–81], the zeta potential for RBC and HeLa is –32 and –19 mV, respectively. In our studies we do not rely on these values since the results obtained by the zeta potential are extremely specific to the microenvironment of the cells, whether they are in a growth medium or in a buffer, and rather characterize the ionic layer around the cell than describe the properties of the cell membrane. In addition, under “natural” conditions, adhesive cultures, HeLa cells are attached to the surface, which limits the possibility of obtaining accurate electrokinetic characteristics since the zeta potential assumes a suspended state of cells, and contact with the surface changes the properties of the cell membrane and charge distribution.
4) The authors should include more detailed information on the correlation between physico-chemical properties of the delivery platforms (size, zeta potential) and the performance on uptake.
Response:
The following fragment was added to the manuscript text:
"Unlike classical PAMAM dendrimers, PAMAM-calix-dendrimers do not exhibit similar patterns associated with their physicochemical properties. This is primarily due to the relatively large size of the thiacalixarene core, which is comparable in linear dimensions to the branches of first-generation dendrimers. This leads to the fact that the nature of uptake, which usually depends on the generation of classical dendrimers, in our case also strongly depends on the core, which makes it problematic to identify relationships only on three generations of compounds. For example, a low level of cellular uptake would be expected for the G1-alt/siRNA complexes at the N:P ratio of 5:1, since the zeta potential of these complexes is barely above zero at this concentration and the hydrodynamic diameter reaches 1 μm. However, even at these ratios, the G1-alt/siRNA complexes have the ability to penetrate into cells at a fairly high level. This appears to be due to a mechanism of entry that appears to be different in the case of G1-alt than other compounds and results in membrane damage. At the same time, the less toxic G2-alt is poorly internalized by cells because the influence of the thiacalixarene core is less significant, and its surface potential is apparently insufficient for cell penetration."
We plan to conduct experiments to determine the mechanism of penetration into cells in future studies with modified PAMAM-calix-dendrimers.